# Changes in the Liver Transplant Waiting List after Expanding to the ‘Up-to-Seven’ Criteria for Hepatocellular Carcinoma

**DOI:** 10.3390/jpm13121670

**Published:** 2023-11-29

**Authors:** Javier Manuel Zamora-Olaya, Ana Aparicio-Serrano, Víctor Amado Torres, Antonio Poyato González, José Luis Montero, Pilar Barrera Baena, Marina Sánchez Frías, Rubén Ciria Bru, Javier Briceño Delgado, Manuel De la Mata, Manuel Rodríguez-Perálvarez

**Affiliations:** 1Division of Hepatology and Liver Transplantation, Reina Sofía University Hospital, 14004 Córdoba, Spain; javierm.zamora.sspa@juntadeandalucia.es (J.M.Z.-O.); ana.aparicio.sspa@juntadeandalucia.es (A.A.-S.); victor.amado.sspa@juntadeandalucia.es (V.A.T.); antonio-poyato.sspa@juntadeandalucia.es (A.P.G.); jluis.montero.sspa@juntadeandalucia.es (J.L.M.); pilar.barrera.sspa@juntadeandalucia.es (P.B.B.); manuel.mata.sspa@juntadeandalucia.es (M.D.l.M.); 2Maimonides Biomedical Research Institute of Cordoba (IMIBIC), 14004 Córdoba, Spain; ruben.ciria.sspa@juntadeandalucia.es (R.C.B.); franciscoj.briceno.sspa@juntadeandalucia.es (J.B.D.); 3CIBER of Hepatic and Digestive Diseases (CIBEREHD), 28029 Madrid, Spain; 4Division of Pathology, Reina Sofía University Hospital, 14004 Córdoba, Spain; marina.sanchez.sspa@juntadeandalucia.es; 5Division of Hepatobiliary Surgery and Liver Transplantation, Reina Sofía University Hospital, 14004 Córdoba, Spain

**Keywords:** liver transplantation, hepatocellular carcinoma, up-to-seven criteria, Milan criteria, waiting list

## Abstract

We aimed to assess changes in the composition of the waiting list for liver transplantation (LT) after expanding from Milan to “up-to-seven” criteria in patients with hepatocellular carcinoma (HCC). A consecutive cohort of 255 LT candidates was stratified in a pre-expansion era (2016–2018; *n* = 149) and a post-expansion era (2019–2021; *n* = 106). The most frequent indication for LT was HCC in both groups (47.7% vs. 43.4%; *p* = 0.5). The proportion of patients exceeding the Milan criteria in the explanted liver was nearly doubled after expansion (12.5% vs. 21.1%; *p* = 0.25). Expanding criteria had no effect in drop-out (12.3% vs. 20.4%; *p* = 0.23) or microvascular invasion rates (37.8% vs. 38.7%; *p* = 0.93). The length on the waiting list did not increase after the expansion (172 days [IQR 74–282] vs. 118 days [IQR 67–251]; *p* = 0.135) and was even shortened in the post-expansion HCC subcohort (181 days [IQR 125–232] vs. 116 days [IQR 74–224]; *p* = 0.04). Tumor recurrence rates were reduced in the post-expansion cohort (15.4% vs. 0%; *p* = 0.012). In conclusion, expanding from Milan to up-to-seven criteria for LT in patients with HCC had no meaningful impact on the waiting list length and composition, thus offering the opportunity for the adoption of more liberal policies in the future.

## 1. Introduction

Hepatocellular carcinoma (HCC) is ranked one of the world’s leading malignancies in terms of incidence and lethality [1]. The first experiences in liver transplantation (LT) to treat HCC were associated with a 43% likelihood of tumor recurrence and 39% overall survival at 5 years, thus placing HCC as a formal contraindication for LT [2]. However, the upcoming Milan criteria, meaning a single nodule ≤ 5 cm or up to 3 nodules ≤ 3 cm each, in the absence of macrovascular invasion or extrahepatic metastases, allowed selecting a subgroup of HCC patients with excellent outcomes after LT [3]. Henceforth, the Milan criteria were considered the standard of care to select HCC candidates for LT [4,5]. Many authors consider that the Milan criteria could be too restrictive, thus precluding access to LT by some patients who would have reduced the risk of tumor recurrence. In recent years, several expansions on the number or diameter of the HCC nodules have shown acceptable tumor-free and overall survival rates after LT [6,7,8,9,10].

There is a general agreement that macrovascular invasion and extrahepatic spreading are formal contraindications for LT [11]. In the absence of these situations, expanding the Milan criteria is parallel to an increase in tumor recurrence rates after LT, motivating the creation of the metro ticket principle: the longer the ride allowed in terms of tumor burden, the higher the price in terms of HCC recurrence rates [7].

The decision to expand the Milan criteria in a certain LT setting requires a careful weighing of risks and benefits. A moderate expansion would allow more patients with HCC to access LT, resulting in better outcomes, particularly considering that LT is the only therapeutic option able to cure both the tumor and the underlying liver disease. On the other hand, a liberal expansion could dramatically increase the number of patients with HCC accessing the waiting list, resulting in a decreased likelihood of transplantation for patients with indications different from HCC.

Spain is a worldwide leader in organ donation and transplantation [12]. With the advent of the direct antivirals against hepatitis C in 2014, a progressive shortening of the waiting list for LT was observed in Spain, with the number of LT candidates from 2015 (*n* = 791) to 2018 (*n* = 386) halved [13]. In this setting, the Spanish Society of Liver Transplantation (SETH), which comprises 25 LT institutions, decided to take advantage of this scenario by promoting in 2019 a national consensus for expanding indications for LT, including HCC. A decision to expand from the Milan criteria to the up-to-seven criteria (i.e., the sum of the number of HCC nodules and the diameter of the large nodules in cm less or equal than seven) [14] was reached, with some restrictions related to serum alpha-fetoprotein (AFP) as follows: serum AFP had to be <400 ng/mL at baseline or decreasing of serum AFP below this threshold after successful local ablation therapy had to be proved [6].

In the present study, we aimed to assess changes to the LT waiting list length and composition after incorporating the “up-to-seven” criteria in patients with HCC.

## 2. Materials and Methods

A consecutive cohort of patients included on the waiting list for LT between January 2016 and August 2021 at our institution was retrospectively included. Exclusion criteria were age < 18 years old, acute liver failure included for urgent transplantation, and combined organ transplantation. The selection of candidates for LT was made by a multidisciplinary team aligning with the guidelines issued by the European Association for the Study of the Liver [4,15]. Imaging and histological criteria to diagnose HCC mirrored current international guidelines [4]. The present study was conducted according to the guidelines of the Declaration of Helsinki (6th revision, 2008) and the protocol was approved by the Andalusian Research Ethics Committee (code UPT07/5746).

Patients were stratified into two groups according to date of inclusion on the waiting list as follows: the pre-expansion cohort comprised patients included from January 2016 to December 2018; and the post-expansion cohort comprised patients included from January 2019 to August 2021. This stratification was performed according to the date when the expansion from the Milan criteria to the up-to-seven criteria was formally adopted in our institution following the recommendations of the Spanish Society of Liver Transplantation (SETH) [6]. The study was performed on an intention-to-treat basis so that all potential candidates for LT entering the waiting list were included irrespective of their indication for LT. This approach allowed analyzing the probability of death on the waiting list or of being excluded because of disease progression before and after the expansion. Any intervention resulting in a significant increase in mortality or delisting due to clinical deterioration in patients with or without HCC should not be considered appropriate. Indications for LT were categorized into three groups: (a) the group of candidates with hepatic insufficiency comprised patients with decompensated cirrhosis and deranged synthetic liver function denoted by coagulopathy, jaundice and/or renal impairment; (b) the group with HCC included those patients with preserved liver function but at least one viable HCC nodule confirmed in dynamic imaging techniques; and (c) the group of special indications included patients with a relatively preserved liver function not meeting criteria of hepatic insufficiency in the blood test analysis who had severe complications of liver disease resulting in an increased risk of dying in the short term or in a derangement in their quality of life. The most prevalent special indications of LT are refractory ascites and chronic/recurrent hepatic encephalopathy, but there are many special indications that may be found elsewhere [15].

In our institution, prioritization on the waiting list was performed using the Model for end-stage Liver Disease (MELD) score [16], which is a formula that combines three objective and readily available analytic parameters: serum creatinine, serum bilirubin and international normalized ratio (INR). The MELD score ranges from 6 to 40 points with an incremental risk of predicted mortality, so the sickest patients with a higher MELD are granted the first positions on the waiting list. Patients with hepatic insufficiency are assigned their true calculated MELD score at listing, and the score may be updated every three months or whenever the patient experiences a significant change in liver function. For indications other than hepatic insufficiency, including patients with HCC, an initial MELD score of 15 points is assigned at listing. Patients with HCC are granted one extra-MELD point on a monthly basis for a maximum score of 23. For patients with special indications other than HCC, one extra MELD point is added every other month, again for a maximum score of 23. This capping ensures that very sick patients with hepatic insufficiency are always prioritized over special indications, including HCC. The only rule exception concerns patients enlisted due to refractory ascites who may benefit from switching to the serum sodium-corrected MELD (MELD-Na) [17] without capping. These prioritization criteria remained unchanged during the study period.

Individuals were followed on the waiting list and eventually after LT until April 2023 to evaluate outcomes. The length of the waiting list was defined as the median time (in days) from the patient’s inclusion on the list to the eventual transplantation or drop-out for any reason. The composition of the waiting list referred to the indication for LT and demographic data of the transplant population. Mortality on the waiting list or exclusion due to tumor progression was compared before and after the implementation of the expansion criteria. In patients with HCC, we analyzed changes in radiological characteristics at baseline (number of HCC nodules and diameter of the largest nodule in millimeters), serum AFP, bridging locoregional ablative therapies while the patient was on the waiting list, and the following histological features of HCC in the explanted liver: number of nodules, diameter of the largest nodule, tumor differentiation according to the Edmonson’s scale [18], and microvascular invasion, defined as the presence of tumor emboli in a portal radicle vein, large capsule vessel or in a vascular space lined by endothelial cells [19]. Finally, we recorded tumor recurrence rates after LT and recurrence-free survival.

After LT, patients were admitted to the intensive care unit until clinical stabilization and, after 4–6 days without complications, they were transferred to the hepatology ward. Patients were usually discharged 8–15 days after surgery and followed in the outpatient clinic weekly within the first month, monthly within the first 6 months, and every 3–6 months thereafter. Patients underwent abdominal doppler ultrasound within the first 24 h after surgery, at day 7, and at least every 4–6 months thereafter. In patients with HCC, serum alpha-fetoprotein was determined in each visit to the outpatient clinic. In HCC patients showing an elevation of serum alpha-fetoprotein or with a suspicious nodule in the liver ultrasound, a whole-body contrast-enhanced computed tomography and a liver magnetic resonance was performed to rule out tumor recurrence.

Categorical variables were presented in absolute numbers and percentages. Continuous variables were expressed as means and standard deviations, except for those with skewed distribution in which the median and the interquartile range (IQR) were used. The appropriate hypothesis contrast tests were used according to the involved variables as follows: chi-square for frequencies, student T for continuous variables with normal distribution and U Mann–Whitney for continuous variables with asymmetric distribution. Kaplan–Meier curves with a log-rank test and univariate/multivariate Cox’s regression were used to analyze time-dependent outcomes such as mortality or delisting due to clinical worsening while the patient is on the waiting list, and tumor recurrence and progression-free survival for patients with HCC accessing LT. Two-sided *p* values were computed and a *p* < 0.05 was considered statistically significant. All analyses were performed by using SPSS 27·0 (IBM, Chicago, IL, USA).

## 3. Results

A total of 255 patients were included in the study: 149 patients in the pre-expansion cohort and 106 patients in the post-expansion cohort. A flowchart of patients accessing LT before and after the expansion of criteria is shown in Figure 1. In the pre-expansion cohort, 87.2% of patients accessed LT compared with 85.8% of patients in the post-expansion cohort (*p* = 0.75). HCC was the most frequent indication for LT in both cohorts, without statistical differences between the pre-expansion era (47.7%) and post-expansion era (43.4%) (*p* = 0.52) [Figure 2]. Again, there were no differences in terms of the likelihood of LT in patients with HCC before and after the expansion (90.1% vs. 84.8%; *p* = 0.38)

There were no differences in terms of the probability of exclusion from the waiting list between the pre- and post-expansion groups (12.8% vs. 14.2%; *p* = 0.75) [Figure 3]. Baseline characteristics of patients with hepatocellular carcinoma included for LT are shown in Table 1. Patients with HCC in the pre-expansion and in the post-expansion periods were comparable in terms of age (57.6 ± 5.4 vs. 58.5 ± 4.9; *p* = 0.38), sex (male 87.3% vs. 87%; *p* = 0.95) and etiology of the underlying liver disease (hepatitis C predominated in both groups: 62% vs. 58.7%; *p* = 0.72). Locoregional treatment was frequently used in both cohorts (89.7% vs. 82.2%; *p* = 0.25). As expected, the proportion of patients with HCC beyond Milan criteria in the pre-transplant radiological assessment was increased in the post-expansion cohort (19.6% vs. 7%; *p* = 0.04). In other words, the proportion of patients who met the Milan criteria in the pre-expansion era was 93% and in the post-expansion era was 80.4%. However, serum alpha-fetoprotein levels were similar in the pre- and post-expansion cohorts (4.9 ng/mL vs. 5.4 ng/mL; *p* = 0.99). 

The overall length on the waiting list was similar in the pre- and post-expansion cohorts (172 days [IQR 74–282] vs. 118 days [IQR 67–251]; *p* = 0.135). Kaplan–Meier curves showing the risk of exclusion from the waiting list in the pre- and post-expansion periods are shown in Figure 4. Individuals included for LT because of hepatic insufficiency or special indications did not suffer significant differences in their length on the waiting list after the expansion criteria were implemented (143.5 days [IQR 27–360] vs. 122 days [IQR 40–310]; *p* = 0.62). On the contrary, among patients with HCC, the median time on the waiting list was significantly shortened in the post-expansion cohort (181 days [IQR 125–232] vs. 116 days [IQR 74–224]; *p* = 0.04). 

The evolution of the waiting list for liver transplantation according to the indication for LT is shown in Figure 5. There was a trend to a progressive decrease in the number of patients active on the waiting list in the most recent years and this trend was consistent for patients with HCC, and also for patients with other indications for LT. This effect was contained in 2020, probably owing to the COVID-19 pandemic, but clearly resumed in 2021. Therefore, the expansion of HCC criteria did not result in more patients awaiting LT.

In the analysis of the explanted liver of patients with HCC, there were no evidence of more advanced tumors in the post-expansion group. No significant differences were found in terms of microvascular invasion rates (37.8% vs. 38.7%; *p* = 0.93), neither in the number of nodules (1.54 ± 0.75 vs. 1.48 ± 0.81; *p* = 0.7), or in the diameter of the largest nodule (28.2 ± 12.7 vs. 28.3 ± 11.2 mm; *p* = 0.97). However, a trend towards an increased number of tumors outside Milan Criteria was observed in the explanted liver in the post-expansion cohort (21% vs. 12.5%; *p* = 0.25).

Overall survival among HCC patients after LT was similar in the pre- and post-expansion eras (80.9% vs. 84.2% at 24 months; *p* = 0.49). There was no documented HCC recurrence in the post-expansion cohort, while in the Milan criteria cohort, there was a 14.1% recurrence rate (log rank *p* = 0.043) (Figure 6). Most cases of HCC recurrence in the pre-expansion cohort occurred within the first year after LT. Recurrence-free survival curves after LT are shown in Figure 7.

In the intention-to-treat analysis, including all candidates for LT, the risk of death while on the waiting list or exclusion due to clinical worsening or tumor progression remained unchanged before and after the expansion to the up-to-seven criteria (*p* = 0.89) (Figure 8).

## 4. Discussion

This is the first study that addressed the effect of the expansion of the Milan criteria in Spain according to the recommendations made by the consensus of the Spanish Society of Liver Transplantation (SETH) [6]. We did not observe a clinically meaningful impact on the waiting list length and composition. In addition, despite that a higher proportion of patients in the post-expansion era were outside the Milan criteria in the explanted liver, the risk of HCC recurrence after LT was even lower.

The Milan criteria are still considered the standard of care to select candidates with HCC for LT [11]. However, there have been many proposals to expand Milan criteria with acceptable tumor recurrence rates and overall survival. In most cases, an increase in the number of nodules, in the size of the nodules or in both features was proposed [20]. The most accepted expansions of Milan criteria, which have been externally validated, are the University of California-San Francisco (UCSF) criteria (a single nodule up to 6.5 cm or up to 3 nodules, less than 4.5 cm each, if the total tumor diameter is less than 8 cm) [21], the criteria from the University of Navarra (a single nodule up to 6.5 cm or up to 3 nodules, less than 5 cm each) [22], and previously referred up-to-seven criteria [14]. There are provocative proposals with more liberal criteria in terms of the higher tumor burden allowed but conditioning the inclusion on the waiting list to patients with low serum alpha-fetoprotein levels, the absence of moderate-poor tumor differentiation in the biopsy, or to patients with successful downstaging to the Milan criteria using locoregional therapies [23]. Among all these options for expanding the Milan criteria, the up-to-seven criteria are considered a moderate expansion, which may not increase abruptly the number of candidates for LT, so it seems reasonable to start expanding with the up-to-seven criteria before considering more liberal policies. Although the “up-to-seven” criteria have already demonstrated similar oncological outcomes as Milan criteria in the absence of microvascular invasion [7,24], it is unclear how expanding HCC transplant criteria would impact on the outcomes of other candidates for LT without HCC. The adoption of more liberal HCC transplant criteria would certainly benefit HCC patients, but if the donor pool is not parallelly increased, there is a risk of penalizing patients with other indications, such as hepatic insufficiency or refractory ascites, ultimately resulting in increased mortality while on the waiting list. In our study, there were no significant differences in the evolution of waiting list composition, which paralleled the trends observed in the whole country. According to data from the *Organización Nacional de Trasplantes* (ONT), the number of LT procedures and the number of patients awaiting LT in Spain have remained unchanged during the study period, with the exception of a transitory decline in donation rates during the initial phase of the Coronavirus Disease 2019 (COVID-19) pandemic [13]. In addition, although a higher tumor burden was accepted, we did not observe increased rates of tumor characteristics related to worse prognosis, such as microvascular invasion, probably owing to additional restrictions regarding alpha-fetoprotein. Indeed, the recommendations of the Spanish Society of Liver Transplantation (SETH) allow up-to-seven criteria as long as strict bounds of serum alpha-fetoprotein are met (i.e., <400 ng/mL at baseline or decreasing of serum AFP below this threshold after successful local ablation therapy). This rule explains why serum alpha-fetoprotein did not increase after the expansion. In any case, it seems that the “up-to-seven” criteria are a mild expansion over Milan criteria, and in light of the present study, more liberal policies could be explored in the future as long as donation rates remain unchanged.

In our study, the radiological assessment of HCC at inclusion on the waiting list showed larger nodules and an increased number of tumors in the post-expansion period. Given that a similar proportion of patients underwent bridging therapies before and after the expansion, and there was even a shorter length on the waiting list for HCC patients in the post-expansion period, more advanced HCC features in the explanted livers of patients transplanted in the post-expansion criteria would be expected. An increase in HCC recurrence rates after LT would be also anticipated. However, this was not the case and tumor recurrence rates were even reduced after the expansion. Although the reduced sample size claims suggest caution when interpreting this reduction in tumor recurrence rates, this could be a potentially relevant observation. The protocol for managing HCC on the waiting list did not differ between the pre- and post-expansion eras, and indeed the use of locoregional ablative therapies was comparable. No adjuvant therapy to prevent HCC recurrence has proven useful hitherto and there was not a significant modification of post-transplant surveillance strategies. One relevant factor to consider is the serum alpha-fetoprotein-derived limitation, which was applied simultaneously with the expansion criteria. More nodules and larger may be allowed within the up-to-seven criteria in the expanded period, but patients with serum alpha-fetoprotein higher than 1000 ng/mL did not enter the waiting list, not even if levels were higher than 400 ng/mL (unless a significant reduction was achieved after locoregional ablation). This could have resulted in a lessened impact on the waiting list length and composition and, at the same time, in the selection of a group of patients for LT with particularly favorable biological HCC behavior. On the other hand, in recent years, the quality of radiological techniques, including magnetic resonance has been significantly improved. New devices provide higher-quality images, which allow detecting more nodules that could have been overlooked in previous years with older equipment. Current technologies could also allow us to better estimate the true diameter of the nodules. This could introduce bias on pure morphological criteria such as Milan or up-to-seven and could buffer the effect of expanding HCC criteria for LT. In this scenario, it is reasonable to further expand HCC criteria for LT beyond up-to-seven, but this decision should not be taken by individual transplant centers, but instead should be based on a nationwide consensus. Apart from expanding morphological criteria, it is paramount to continue improving our understanding of tumor biology to incorporate novel biomarkers such as cell-free DNA or circulating tumor cells in the decision-making process [25].

HCC downstaging may not have played a role in patient selection in this cohort as this strategy did not form part of routine practice in our center during the study period. In contrast, most patients underwent bridging locoregional therapy, mostly trans-arterial chemoembolization, to decrease the risk of tumor progression and drop-out. Tumor progression after locoregional ablative therapies should motivate exclusion from the waiting list and this criterion remained after expanding from Milan criteria to up-to-seven criteria to avoid patients with biologically aggressive tumors accessing LT with an increased likelihood of post-LT tumor recurrence [26,27,28]. If a future expansion over the up-to-seven criteria is considered in Spain, a progressive increase in serum AFP and tumor progression on the waiting list despite locoregional ablative therapies should be kept as red flags to preclude LT [29,30].

The present study is limited by its retrospective design and single center involvement. However, our LT institution traditionally holds one of the longest waiting lists for LT in Spain and, therefore, would have been more vulnerable to the effect of expanding indications in HCC, making our results of high interest for other centers considering the expansion of HCC transplant criteria. In addition, our results may have been influenced by other clinical practice changes implemented during the study period, including the transitory donor shortening observed during the COVID-19 pandemic in 2020.

## 5. Conclusions

The expansion of HCC criteria for LT from the Milan criteria to the up-to-seven criteria, aligning with the Spanish Society of Liver Transplantation (SETH) consensus, had no meaningful impact on the waiting list length and composition. In addition, tumor recurrence rates after LT remained stable, or even decreased, after the expansion. Should the current trends in donation continue, a more liberal policy for selecting LT candidates with HCC could be considered, although continuous monitoring of waiting list and transplant outcomes will be required.

## Figures and Tables

**Figure 1 jpm-13-01670-f001:**
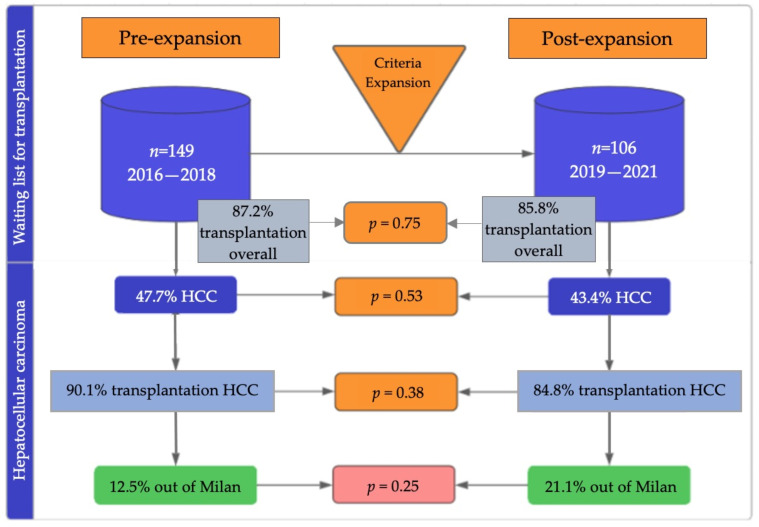
Flowchart of patients accessing liver transplantation before and after the expansion of criteria from Milan to up-to-seven. Patients included before expanding criteria are shown on the left and patients included after the expansion are shown on the right. The probabilities of accessing transplantation in each group, including all indications for liver transplantation are shown in the upper part of the figure, while the probabilities of accessing transplantation in patients with hepatocellular carcinoma are shown in the lower part of the figure. The accessibility to liver transplantation was similar before and after the expansion, both in the overall cohort and in the subgroup with hepatocellular carcinoma. HCC: hepatocellular carcinoma.

**Figure 2 jpm-13-01670-f002:**
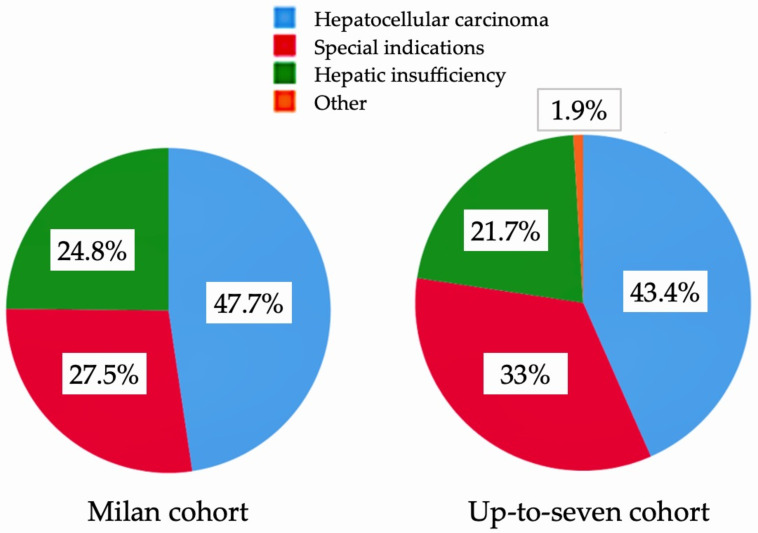
Indications for liver transplantation. The implementation of expanded criteria for hepatocellular carcinoma did not change the composition of the waiting list.

**Figure 3 jpm-13-01670-f003:**
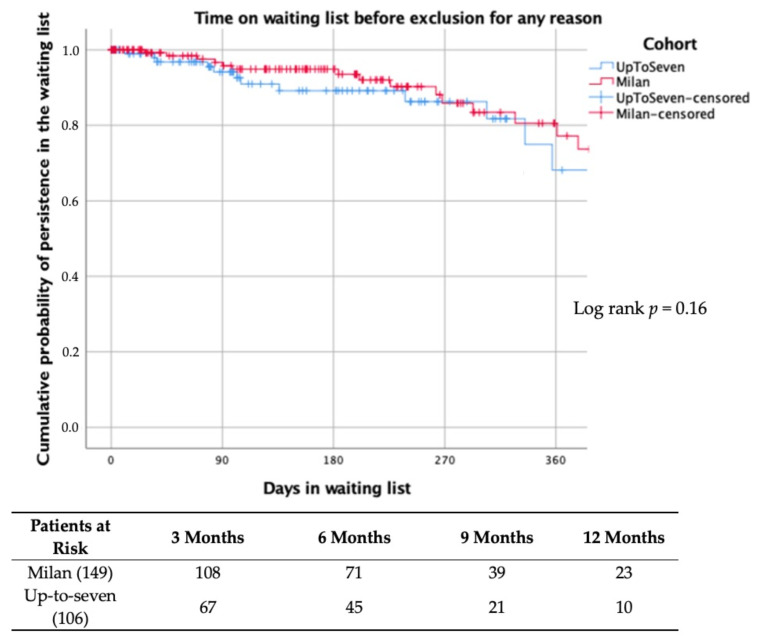
Kaplan–Meier curve showing the cumulative risk of death or exclusion from the waiting list for any reason. Patients were censored a transplantation for this analysis.

**Figure 4 jpm-13-01670-f004:**
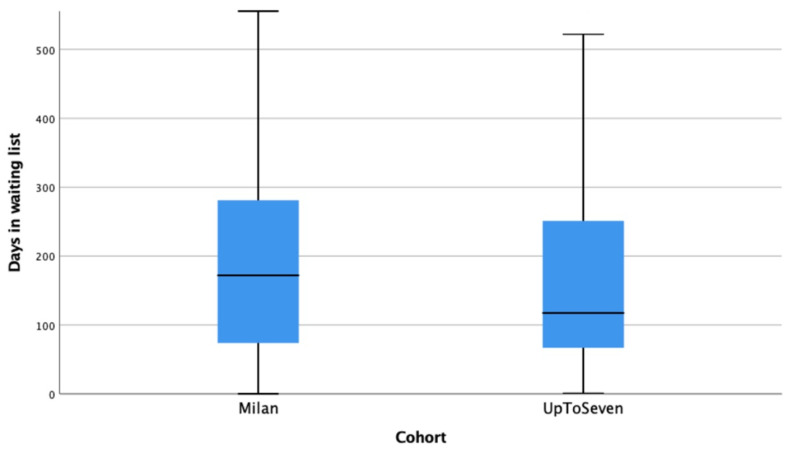
Median time on the waiting list for individuals within each comparison group.

**Figure 5 jpm-13-01670-f005:**
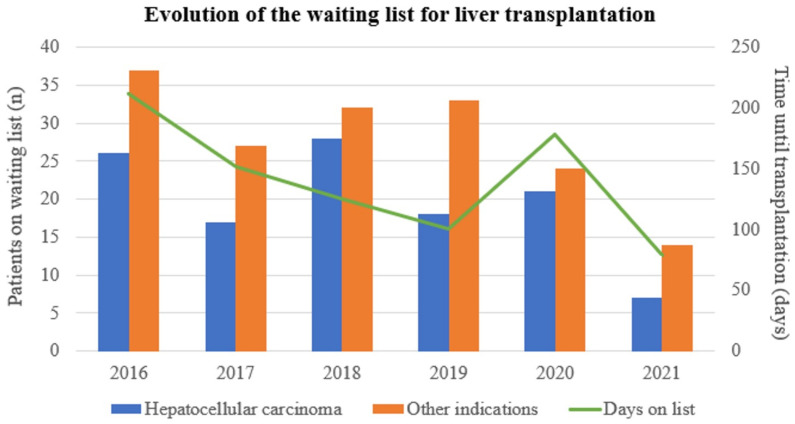
Absolute number of patients active on the waiting list at the end of each year of the study period disaggregated by the indication of liver transplantation (bars). The evolution of the median time on the waiting list is presented as a continuous line.

**Figure 6 jpm-13-01670-f006:**
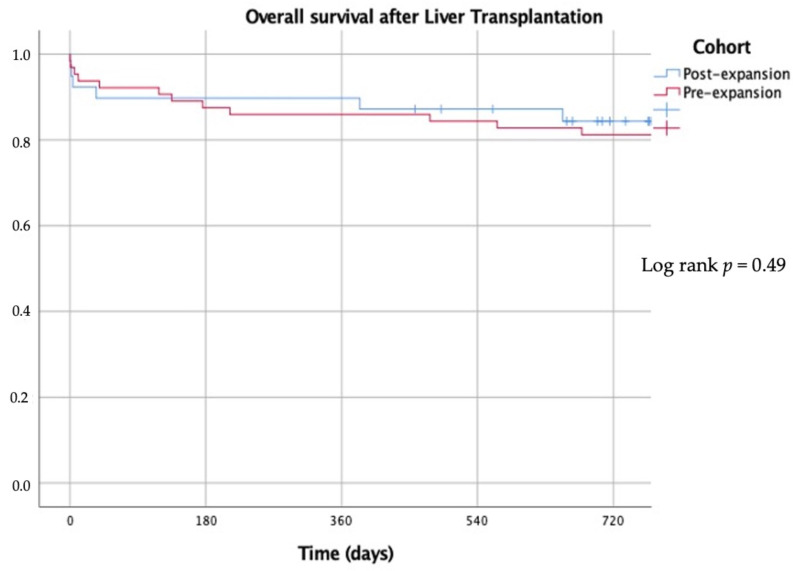
Overall survival after liver transplantation in patients with hepatocellular carcinoma.

**Figure 7 jpm-13-01670-f007:**
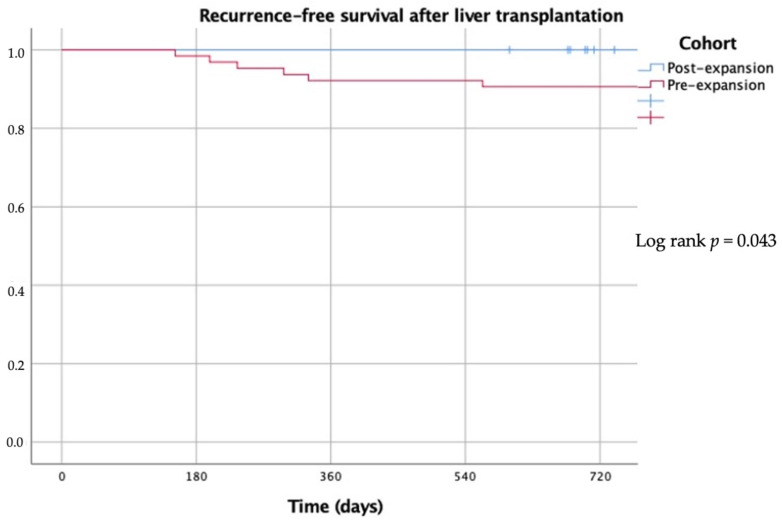
Kaplan–Meier curves showing recurrence-free survival after liver transplantation in patients with hepatocellular carcinoma.

**Figure 8 jpm-13-01670-f008:**
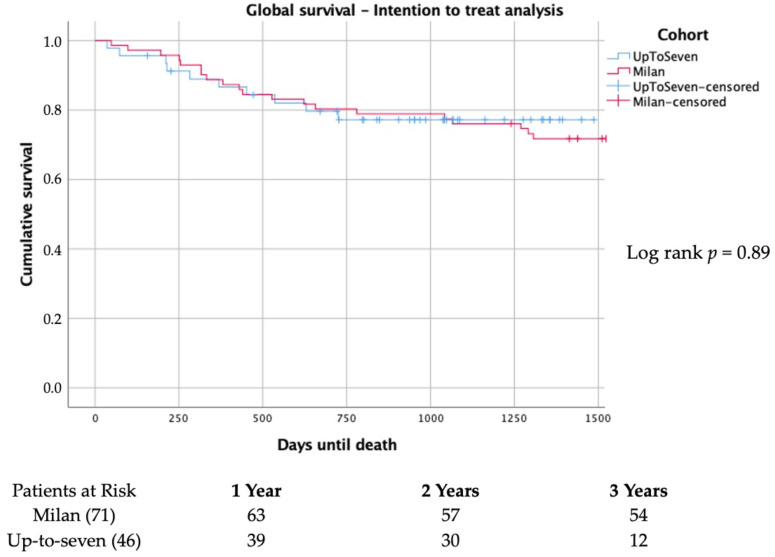
Intention to treat analysis on overall survival in patients with hepatocellular carcinoma accessing the waiting list for liver transplantation.

**Table 1 jpm-13-01670-t001:** Baseline characteristics of patients with hepatocellular carcinoma included for liver transplantation in the pre-expansion cohort (*n* = 71) and post-expansion cohort (*n* = 46).

	Pre-Expansion Period (2016–2018)	Post-Expansion Period (2019–2021)	*p*
Age (years)	57.6 ± 5.4	58.5 ± 4.9	0.38
Sex (men); *n* (%)	62 (87.3%)	40 (87%)	0.95
Hepatitis B; *n* (%)	12 (16.9%)	3 (6.5%)	0.1
Hepatitis C; *n* (%)	44 (62%)	27 (58.7%)	0.72
Alcoholic liver disease; *n* (%)	33 (46.5%)	26 (56.5%)	0.29
Portal Hypertension; *n* (%)	63 (88.7%)	42 (89.1%)	0.95
Child-Pugh score	5 (IQR 5–7)	6 (IQR 5–7)	0.49
MELD score	9 (IQR 7–12)	9 (IQR 8–13)	0.17
Milan criteria out at inclusion (radiological), *n* (%)	5 (7%)	9 (19.6%)	0.04
Number of nodules in the explanted liver, *n* (%) Single nodule Multinodular	42 (59.2%) 29 (40.8%)	31 (67.4%) 15 (32.6%)	0.37
Main nodule diameter in the explanted liver (mm)	28.2 ± 12.7	28.3 ± 11.2	0.97
Histological differentiation (moderate-poor); *n* (%)	24 (58.5%)	12 (48%)	0.47
Microvascular invasion; *n* (%)	17 (37.8%)	12 (38.7%)	0.93
Baseline AFP (ng/mL)	4.9 (IQR 3.2–26)	5.4 (IQR 3–17)	0.99
Bridging locoregional therapy; *n* (%)	61 (89.7%)	37 (82.2%)	0.25
Milan criteria out in the explanted liver, *n* (%)	9 (12.5%)	10 (21.1%)	0.25

## Data Availability

The deidentified raw database is accessible in a public Mendeley repository. Please contact the authors for more details.

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
