# Peer review of "Changes in the Liver Transplant Waiting List after Expanding to the ‘Up-to-Seven’ Criteria for Hepatocellular Carcinoma"

_jpm, 2023, doi:10.3390/jpm13121670_

Round 1
Reviewer 1 Report
Comments and Suggestions for Authors
The study by Zamora-Olaya aimed to assess the impact of expanding liver transplantation criteria for HCC patients from Milan to up-to-seven criteria. The authors compared two cohorts before and after the expansion and found that the waiting list composition and length remained largely unchanged. Tumor recurrence rates decreased in the post-expansion cohort. Overall, the study supports the feasibility of more liberal transplantation criteria for HCC patients. However, while the up-to-seven criteria have been proposed and employed for more than a decade, this study does not contribute significantly to the existing knowledge in this area. Furthermore, substantial revisions are recommended to enhance the manuscript's quality.
1. The author should provide a comprehensive overview of the allocation criteria for HCC patients in their specific country. It is important to clarify whether HCC patients receive additional scores based on the Model for End-Stage Liver Disease (MELD) score, and if so, how this scoring system impacts their prioritization for liver transplantation. This information is crucial for readers to understand the context and nuances of HCC patient selection and allocation in the studied region.
2. It is interesting that a consensus on the use of up-to-seven criteria in Spain was only reached in 2019, despite the proposal of these criteria many years prior. It would be valuable for the author to elaborate on how HCC patients were listed and prioritized in transplantation centers before 2019. Were Milan criteria the sole criteria used during this period, or were there variations in practice across different centers?
3. In the post-expansion cohort of 106 patients, it would be beneficial for the study to provide a more detailed breakdown of how many of these patients still met the Milan criteria. A further stratification between patients who met Milan criteria and those who did not would help to eliminate potential biases and provide a clearer understanding of the impact of the expansion criteria on these subgroups.
4. Figure 1 presents some confusion. Given that this study primarily focuses on HCC patients, it raises questions regarding why the indication for HCC in both groups is less than 50%. Additionally, it's unclear what the figures 90.1% and 84.8% represent in terms of transplantation. To ensure clarity, the author should provide a more detailed explanation of the data, including the percentages and their relevance to the study's objectives.
5. In Figure 2, it would be helpful for the author to provide specific details regarding "hepatic insufficiency" and elucidate what falls under the category of "special indications," as they account for a substantial proportion.
6. The observation that patients in the post-expansion era had a significantly lower risk of HCC recurrence is intriguing and warrants further discussion. It would be valuable for the author to delve into potential reasons for this difference. Do treatment strategies differ between the pre- and post-expansion eras? Exploring potential changes in treatment protocols, patient selection criteria, or advancements in HCC management could help shed light on the reduced recurrence risk observed in the post-expansion cohort.
7. The description of post-LT treatment strategy of HCC patients in author’s center is appreciated.
Comments on the Quality of English Languagemoderate editing
Author Response
#TO THE REVIEWER 1:
* Original comment 1: “The author should provide a comprehensive overview of the allocation criteria for HCC patients in their specific country. It is important to clarify whether HCC patients receive additional scores based on the Model for End-Stage Liver Disease (MELD) score, and if so, how this scoring system impacts their prioritization for liver transplantation. This information is crucial for readers to understand the context and nuances of HCC patient selection and allocation in the studied region.”
Answer: The reviewer is right. The allocation criteria and prioritization systems for patients with hepatocellular carcinoma are heterogeneous and their description is crucial to interpret the results of the study. We have included a dedicated paragraph in the methods’ section as follows: “In our institution, prioritization in the waiting list was performed using the Model for End-stage Liver Disease (MELD) score, which is a formula that combines three objective and readily available analytic parameters: serum creatinine, serum bilirubin and international normalized ratio (INR). The MELD score ranges from 6 to 40 points with an incremental risk of predicted mortality so the sickest patients are granted the first positions in the waiting list. Patients with hepatic insufficiency are assigned their true calculated MELD score at listing, and the calculation may be updated every three months or whenever the patient experiences a significant change in the liver function. For indications other than hepatic insufficiency, including patients with HCC, an initial MELD score of 15 points is assigned at listing. Patients with HCC are granted one extra-MELD point in a monthly basis for a maximum score of 23. For patients with special indications other than HCC, one extra MELD point is added every other month, again for a maximum score of 23. This capping ensures that very sick patients with hepatic insufficiency are always prioritized over special indications, including HCC. The only rule exception concerns to patients enlisted due to refractory ascites who may benefit from switching to the serum sodium-corrected MELD (MELD-Na) without capping. These prioritization criteria remained unchanged during the study period”.
* Original comment 2: “It is interesting that a consensus on the use of up-to-seven criteria in Spain was only reached in 2019, despite the proposal of these criteria many years prior. It would be valuable for the author to elaborate on how HCC patients were listed and prioritized in transplantation centers before 2019. Were Milan criteria the sole criteria used during this period, or were there variations in practice across different centers?”
Answer: We agree with the reviewer that some extra background is needed to understand the rationale of the study and to avoid misunderstanding. In the study period before 2019, we strictly applied Milan criteria as stated in methods without variations. There have been many proposals to expand Milan criteria during the last two decades with a variable extent in terms of number and size of the nodules allowed. Some of these models implemented serum alpha-fetoprotein in combination with morphological features. The implementation of expanded HCC criteria (ie. over Milan criteria) depends on your waiting list and the possibility to expand the donor pool. In Spain, as well as in other countries, the upcoming of the new direct antivirals against hepatitis C resulted in a significant shortening of the transplant waiting list. Between 2015 and 2018 the number of patients awaiting liver transplantation in Spain was halved and this created the appropriate situation to consider the expansion of Milan criteria. In our national consensus, we opted for the “Up-to-seven criteria” including some serum alpha-fetoprotein restrictions because this was considered a moderate expansion which would not create a drastic impact in the waiting list composition. However, our study has shown that this expansion was probably too cautious since we did not notice any impact on the waiting list. We strongly believe that more liberal expansions could be implemented in the future. We have provided a detailed explanations about the different options to expand Milan criteria and the risk/benefit balance of this decision both in the introduction and in the discussion sections.
* Original comment 3: “In the post-expansion cohort of 106 patients, it would be beneficial for the study to provide a more detailed breakdown of how many of these patients still met the Milan criteria. A further stratification between patients who met Milan criteria and those who did not would help to eliminate potential biases and provide a clearer understanding of the impact of the expansion criteria on these subgroups.”
Answer: This information was available in the results’ section of the previous version of the manuscript, but we have given more details for improved clarity. The statement reads as follows: “The proportion of patients with HCC beyond Milan criteria in the pre-transplant radio-logical assessment was increased in the post-expansion cohort (19,6% vs. 7%; p=0.04). In other words, the proportion of patients who met Milan criteria in the pre-expansion era was 93% and in the post-expansion era was 80.4%.”
* Original comment 4: “Figure 1 presents some confusion. Given that this study primarily focuses on HCC patients, it raises questions regarding why the indication for HCC in both groups is less than 50%. Additionally, it's unclear what the figures 90.1% and 84.8% represent in terms of transplantation. To ensure clarity, the author should provide a more detailed explanation of the data, including the percentages and their relevance to the study's objectives.”
Answer: We would like to note that we conceived this study as an intention-to-treat analysis of all potential outcomes in the waiting list. Therefore, instead of focusing only on patients with HCC receiving a liver transplant, we considered all patients who were admitted in the waiting list irrespective of the indication for liver transplantation. This approach allowed to analyze the probability of death in the waiting list or of being excluded from the waiting list before and after the expansion. If an expansion of Milan criteria resulted in a significant increase of mortality or delisting due to clinical deterioration in patients with or without HCC, the expansion should not be considered appropriate. This clarification was added in methods. Fortunately, this was not the case and figure 1 illustrates the probability of receiving a liver transplantation before and after the expansion, which was broadly similar, both in patients with HCC and in patients with indications for liver transplantation other than HCC. We believe that the figure is adequate, but we have provided more detailed information in the text and in the figure legend to allow a better understanding of its content.
* Original comment 5: “In Figure 2, it would be helpful for the author to provide specific details regarding "hepatic insufficiency" and elucidate what falls under the category of "special indications," as they account for a substantial proportion.”
Answer: In methods, we have provided detailed explanations of each potential indication of liver transplantation. Briefly, hepatic insufficiency pertains to patients with decompensated liver cirrhosis who have a deranged liver function defined as a MELD score greater than 15 points. Patients with special indications are those whose MELD score is lower, but experienced severe complications of the liver disease resulting in an increased risk of dying in the short term or in a derangement in their quality of life. The most frequent indications in this miscellanea group are recurrent hepatic encephalopathy and refractory ascites. We have included also detailed information on the prioritization system used in our setting in response to the reviewer’s comment #1, which remained unchanged over the study period.
* Original comment 6: “The observation that patients in the post-expansion era had a significantly lower risk of HCC recurrence is intriguing and warrants further discussion. It would be valuable for the author to delve into potential reasons for this difference. Do treatment strategies differ between the pre- and post-expansion eras? Exploring potential changes in treatment protocols, patient selection criteria, or advancements in HCC management could help shed light on the reduced recurrence risk observed in the post-expansion cohort.”
Answer: Thank you for your interesting insight. Treatment and follow-up strategies before and after expansion did not differ as we have stated in the method’s section of the new version of the manuscript. We have expanded the discussion trying to unveil this issue. The restriction pertaining serum alpha-fetoprotein for patients with morphologically expanded criteria may have played a role. Indeed, we have allowed patients with increased tumor burden to access liver transplantation but only those with serum AFP<400 according to the Spanish consensus document. Therefore, patients with an aggressive biological behavior of HCC were precluded to enter the waiting list, thus reducing the probability of tumor recurrence after liver transplantation. Although we cannot make solid conclusions a further expansion of morphological criteria (beyond Up-to-seven) could be attempted safely if the AFP derived restrictions are kept.
* Original comment 7: “The description of post-LT treatment strategy of HCC patients in author’s center is appreciated.”
Answer: Thank you for your comment. We have provided detailed information in methods.
Reviewer 2 Report
Comments and Suggestions for Authors
The manuscript is well-written, and the study is carefully designed. The topic is highly relevant nowadays. For almost three decades, the so-called Milan criteria have been the gold standard for enrolling in the transplant waiting list for orthotopic liver transplantation of a patient with hepatocellular carcinoma. Many groups in different countries have reported good outcomes despite patient selection beyond the Milan criteria (for tumor size and number).
Therefore, this manuscript is not delivering new information but is a confirmatory study that I consider helpful for liver transplant surgeons and hepatologists.
The references and tables are appropriate. The conclusion is consistent with the evidence presented by the Authors.
Author Response
#TO THE REVIEWER 2:
* Original comment: “The manuscript is well-written, and the study is carefully designed. The topic is highly relevant nowadays. For almost three decades, the so-called Milan criteria have been the gold standard for enrolling in the transplant waiting list for orthotopic liver transplantation of a patient with hepatocellular carcinoma. Many groups in different countries have reported good outcomes despite patient selection beyond the Milan criteria (for tumor size and number). Therefore, this manuscript is not delivering new information but is a confirmatory study that I consider helpful for liver transplant surgeons and hepatologists. The references and tables are appropriate. The conclusion is consistent with the evidence presented by the Authors.”
Answer: Thank you for your kind and positive evaluation. We hope that our results will encourage other transplant teams to expand HCC criteria for liver transplantation so more patients could benefit of this curative therapeutic option.
Round 2
Reviewer 1 Report
Comments and Suggestions for Authors
My questions have been well addressed.